# SHARE OR SPLIT : WHICH IS MORE EFFICIENT?

**Yinpeng Chen, Zicheng Liu, Lijuan Wang & Zhengyou Zhang**
Microsoft Research
Redmond, WA 98052, USA
{yiche, zliu, lijuanw, zhang}@microsoft.com

## ABSTRACT

In this paper, we investigate two different feature representations in convolutional neural networks (CNN): (a) *Share* - all classes share the same feature space, and a fully connected layer is used to decode class activation, and (b) *Split* - each class has its own feature space and a class is activated if its corresponding feature vector has a large norm. This is inspired by Capsules (Sabour et al. (2017)) which splits the feature space. We compare these two representations on a transformed MNIST dataset, which adds random scales and translations on the original digits. The experimental results show that *Split* has better performance when data is limited, while *Share* is better when data is big.

## 1 INTRODUCTION

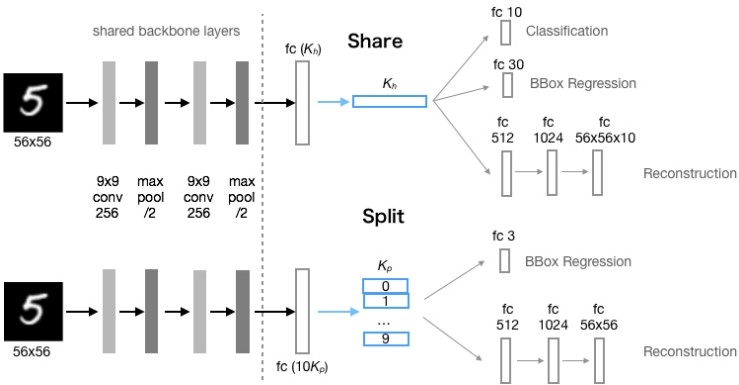

Figure 1: Networks for both *share* and *split* on transformed MNIST dataset.

In the recently published Capsules (Sabour et al. (2017)), each class has its own vector representation (named capsule) that is connected to low level capsules by dynamic routing. Hence, different classes do not share feature space. This is very different from the existing CNN based image classification (LeCun et al. (1998a), Krizhevsky et al. (2012), Simonyan & Zisserman (2015), Szegedy et al. (2015), He et al. (2016)) or object detection (Girshick (2015), Ren et al. (2015), Redmon et al. (2016), He et al. (2017)), in which all classes share the same feature space. The class label and bounding box are decoded from the feature vector by a fully connect layer. *Sharing* or *Splitting* the feature space, which is more efficient under what circumstances?

In this paper, we focus on comparing the two representations, *Sharing* or *Splitting* the feature space, within the scope of CNN. Although this is inspired by capsules, this paper does *not* compare capsules with CNNs. The comparison between *Share* and *Split* is performed on both single task (classification alone) and multi-tasks (classification and bounding box regression) on a transformed MNIST dataset. We found that (a) on classification task alone, *Share* is better when we have enough data, while *Split* is more effective when data is limited, (b) *Share* outperforms *Split* on joint classification and detection.

## 2    SHARING OR SPLITTING FEATURES

The neural networks for classification can be viewed as a process of encoding and decoding. An input image $\boldsymbol{x}$ is encoded to a feature vector $\boldsymbol{f} = E(\boldsymbol{x})$, and then decoded to probabilities over $N$ classes $\boldsymbol{y} = [y_1, \ldots, y_N] = D(\boldsymbol{f})$. This paper focuses on the feature representation $\boldsymbol{f}$ and the decoder $D(\boldsymbol{f})$. We compare two different representations: *sharing* features and *splitting* features.

**Sharing features**: most of the CNNs (LeCun et al. (1998a), Krizhevsky et al. (2012), Simonyan & Zisserman (2015), Szegedy et al. (2015), He et al. (2016)) for classification are in the *share* category. A fully connected layer is used as a decoder $D(\boldsymbol{f})$ to predict the class label. Let us denote the length of the feature vector $\boldsymbol{f}$ as $K_h$. All classes share the $K_h$ feature channels, which is efficient to encode class contexture information. However, it introduces complexity on the decoding side - a fully connected layer is needed to decode the class label.

**Splitting features**: in the recent published Capsules (Sabour et al. (2017)), each class locks its own feature channels (named capsule), which are not shared with other classes. Thus, the feature can be represented as $\boldsymbol{f} = [\boldsymbol{f}_1, \ldots, \boldsymbol{f}_N]$, where $\boldsymbol{f}_k$ is the sub-feature vector for the $k^{th}$ class. All sub feature vectors have equal length $K_p$. The decoder for the splitting features is simple: the probability of each class is proportional to the norm of its corresponding feature vector ($y_k \propto \|\boldsymbol{f}_k\|$). Although the splitting feature representation is less efficient since classes may share common feature channels, its decoder is simpler with zero parameters.

**Which one is better?** *Sharing* and *Splitting* have different tradeoffs between compression and complexity. *Sharing* is efficient on compressing features with a complex decoder, while *Splitting* has redundant representation but a simple decoder. Our conjecture is the preference is related to the size of training data. *Sharing* may leverage the power of big data to learn the parameters in the decoder, while *Splitting* is more effective when data is limited due to its simple decoder. The results are shown in the next section.

Figure 1 shows an example of sharing features and splitting features on hand written digit classification (10 classes). Both networks share the same backbone layers (two convolution and two max pooling layers). The *Sharing* network generates the feature vector by a fully connected layer with $K_h$ outputs and then decodes class labels and bounding box. Since the bounding box is square for MNIST, it is represented by three parameters (one for scale and two for translation). The *Splitting* network generates the feature vector by a fully connected layer with $10K_p$ outputs, and then splits it to 10 sub vectors with length $K_p$ for their corresponding digits. Each sub vector is normalized by using squash in (Sabour et al. (2017)). We use the same reconstruction structure in (Sabour et al. (2017))) as regularization. In the *Sharing* network, we combine cross entropy classification loss, $L_2$ bounding box regression loss and $L_2$ reconstruction loss. In the *Splitting* network, we follow (Sabour et al. (2017)) to use marginal classification loss, $L_2$ bounding box regression loss and $L_2$ reconstruction loss. The weight for reconstruction loss is set to 0.0005.

## 3    EXPERIMENTAL RESULTS

In this section, we discuss the experimental results for both *Share* and *Split* on a transformed MNIST dataset, which transformed the original MNIST dataset (LeCun et al. (1998b)) by adding random scales and translations.

### 3.1    TRANSFORMED MNIST DATASET

We generate two Transformed MNIST training datasets and one test dataset. All transformed images have bigger size $56 \times 56$. A transformed image is generated by three steps: (a) choose an image from MNIST, (b) scale it by a random factor between 0.5 to 2.0, and (c) randomly place it within the $56 \times 56$ bounding box.

The test dataset has 5000 test images, transformed from 5000 MNIST test images. The first training dataset (referred to **Train55K**) has 55000 images by transforming each MNIST training image once. The second training dataset (referred to **Train4.4M**) has 4.4M images, by transforming each MNIST training image 80 times with different scales and translations. Train4.4M can be considered as data augmentation.

| | Classification alone | | Classification and BBox Regression | | | |
|---|---|---|---|---|---|---|
| | 55K | 4.4M | Cls-55K | Cls-4.4M | BBox-55K | BBox-4.4M |
| Share $K_h = 40$ | 98.15 | **99.35** | 98.05 | 99.31 | 85.47 | 82.54 |
| Share $K_h = 80$ | 98.12 | 99.24 | 98.11 | 99.34 | **87.02** | 83.24 |
| Share $K_h = 120$ | 97.97 | 99.33 | 98.12 | **99.35** | 86.16 | 83.40 |
| Share $K_h = 160$ | 97.98 | 99.33 | 98.12 | 99.30 | 86.02 | **83.92** |
| Split $K_p = 8$ | **98.41** | 99.11 | 98.16 | 99.16 | 77.61 | 80.94 |
| Split $K_p = 16$ | 98.36 | 99.11 | 98.11 | 99.26 | 78.95 | 81.94 |
| Split $K_p = 24$ | 98.11 | 99.06 | 98.15 | 99.23 | 79.76 | 82.83 |
| Split $K_p = 32$ | 98.06 | 99.16 | **98.23** | 99.17 | 80.10 | 82.66 |

Table 1: Classification accuracy and bounding box IoU of *Share* and *Split* networks on the Transformed MNST dataset

## 3.2 IMPLEMENTATION DETAILS

The *Sharing* network and the *Splitting* network are shown in Figure 1. Each network is trained on both Train55K and Train4.4M and tested on 5000 transformed test images. We perform the experiment on the *Sharing* network with four different feature lengths: $K_h = 40, 80, 120, 160$ and on the *Splitting* network with four different sub-feature lengths: $K_p = 8, 16, 24, 32$. The weight for reconstruction is set to 0.0005. The mini-batch size is set to 64. We use an Adam optimizer with an initial learning rate of $10^{-3}$. When training on Train55K, we shrink the learning rate by a factor of 0.1 at 60 epochs and finish the training after 80 epochs. Training on Train4.4M only has one epoch, which has the same number of mini-batches as Train55K. The learning rate is reduced by a factor of 0.1 after training 3.3M images.

## 3.3 RESULTS

We run experiments on two tasks: (a) classification alone, and (b) classification and bounding box regression. For classification alone, the bound box regression branch is removed from Figure 1. All experimental results are shown in Table 1.

**Classification Alone**: The results validate our conjecture. The *Split* network performs better on Train55K, while the *Share* network is more accurate on Train4.4M. This is because that the simpler decoder is helpful to prevent overfitting. We observe overfitting on both *Share* and *Split*, since the accuracy decreases as the feature dimension increases.

**Classification and Bounding Box Regression**: We have two major observations as follows:

*Observation 1*: On Train55K, *Share* is slightly behind *Split* on classification accuracy, but its bounding box IoU outperforms *Split* by a big margin. On Training4.4M, *Share* is better on both classification and bounding box regression. This is because the bounding box regression is highly correlated across different digits and this correlation is not leveraged when feature is split per class.

*Observation 2*: From Train55K to Train4.4M, *Split* has better performance on both classification and bounding box regression due to the power of data augmentation. However, even though *Share* has better classification results on Train4.4M, its bounding box IoU drops about 3 points compared to Train55K. The different trend between *Share* and *Split* may be due to two reasons. Firstly, on Train55K, the bounding box regression is not well trained for *Split*, but is well trained for *Share*. When splitting the feature, each digit learns its own way to regress bounding boxes and the relationship between digit pixels and bounding box across different digits is ignored. Secondly, fitting the classification on Train4.4M is significantly harder than Train55K, which may sacrifice the performance of bounding box regression. We conjecture that the combination of these two factors causes performance gain on bounding box IoU for *Split* and performance loss on bounding box for *Share*.

**Take Home Messages**: our experimental results recommend using *Split* on classification task alone when data is limited and using *Share* when big data is available. When dealing with joint classification and detection, *Share* is recommended.

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
