# OpenReview forum: "Share or Split : which is more efficient?"
_ICLR.cc/2018/Workshop — Reject_

### Official Review · AnonReviewer1 · 2018-03-09
**hard to draw many conclusions**

**Rating:** 5
**Confidence:** 3

**Review:**

This paper compares two approaches to final feature layer representation in a classifier:  "Share", which uses the same feature vector for all classes, and "Split", which has a different feature vector for each class, and chooses the "winner" based on which one has largest magnitude.  These two representations are used to classify transformed MNIST digits, as well as estimate digit location on a black background with a bounding box regressor.

Unfortunately, it's hard for me to figure out exactly what to take away from this.  The conclusion at the end says "our experimental results recommend using Split on classification task alone when data is limited and using Share when big data is available. When dealing with joint classification and detection, Share is recommended".  But it seems a stretch to draw even this conclusion:  Share seems better in 3/4 settings, and in the one where it performs worse, it is by a small margin --- about 98.05% vs 98.25%.  So it's hard to know if this is indeed from the architecture structure, or from other optimization, loss or parameter differences.  I'm also not sure how many trials were performed or what is the difference in error between trials -- it appears it would be somewhere on the order of 0.1% since K_h=80 is about 0.1% worse than K_h=40 for 4.4M dataset, when one would expect it to be better with this much data.

Since many of these classification results are close, I wonder whether it is possible to draw any equivalences between the two models?  For instance, if L1 norm were used instead of L2, this could be computed by adding an additional layer on top of the hidden representation with a block structure that has 1's in locations that correspond to the entries with sums in the L1.  In that case, the differences are that the top layer is fixed to block sums, versus learned, and the loss used.

The clearest conclusion appears to be that Split is worse for bounding box regression on transformed MNIST, since the task is roughly the same for every class, meaning that sharing the features between classes should have a significant benefit here --- which it does, as the authors point out.

Beyond this, however, it seems hard to draw many conclusions from this experiment.  The difference in error is pretty small, and in a single experimental setting.

---

### Official Review · AnonReviewer2 · 2018-03-10
**-**

**Rating:** 5
**Confidence:** 4

**Review:**

This paper is a straightforward comparison of convnets trained using conventional multinomial logistic regression-based classification (“sharing”) vs. the recently proposed capsules (“splitting”).  Both methods are evaluated on “transformed MNIST” for classification and bounding box regression using different feature dimensionalities in the last hidden layer as well as different training set sizes.  Based on the results, the paper concludes that “sharing” works better with sufficiently large training sets, but “splitting” is better with smaller training sets.

Given that the paper is essentially evaluating two existing methods, this is slightly below threshold for me as it seems a bit too limited to draw any robust conclusions from.  Detailed comments:

- An L2 reconstruction loss is used to train both networks.  It would have been nice to know what the effect of this loss term on classification / bbox regression accuracy is, as well as the reconstruction error of each network.

- The splitting network is reported as being significantly worse for bbox regression (with far larger performance differences than for classification), but I suspect this is due to a particular design choice: in the “splitting” network the bounding box regression and reconstruction networks take the “split” features as input.  This is potentially detrimental as the split features must now be both class-specific (as their norm is used directly for classification) but also must contain class-generic information used for bbox regression and reconstruction.  An alternative that might make for a fairer comparison with the sharing network would be to use the “split” features for classification alone and then have a separate output branch used for bbox regression and reconstruction.  (An equivalent design could be used for the “sharing” network to keep the capacity comparable.)

- The networks are only evaluated on an MNIST-derived dataset.  Given this limited evaluation and the quite small differences in classification accuracy between all the methods, I would have liked to see some measure of variance (e.g. confidence intervals) over difference dataset splits and random weight initializations in the results.  Furthermore, it’s not clear that the conclusions would be similar on larger or more visually complex datasets like ImageNet or even CIFAR.

---

### Official Review · AnonReviewer3 · 2018-03-11
**Lack of experiments**

**Rating:** 5
**Confidence:** 4

**Review:**

This paper investigates two different feature representations: share and split. Share is the conventional way for image classification, i.e. all classes share the same feature space and a fully connected layer determines the class activation. Split is another way by using separate feature space for each class and activation is determined by which class having a large norm. Experiments on transformed MNIST dataset show that Split is better for limited data and Share is better for enough data.

Strength and weakness
+ Explanation of pros and cons.
- Results only on one dataset.
- The backbone network is rather simple. It would be great to try deeper network to see if more powerful encoder would have different result.
- For dataset sizes, there are only two options Train 55k and Train 4.4M. Also difference of number in Table 1 is very small, not sure if it is marginally large enough to convince the take away message.

In general, I think this is a very interesting topic. However, experimental results are only on one dataset and the numbers are not significantly enough for the statement.

---

### Decision · Program_Chairs · 2018-03-20
**ICLR 2018 Workshop Acceptance Decision**

**Decision:**

Reject

**Comment:**

Based on the reviews, this paper has not been accepted for presentation at the ICLR workshop. However, the conversation and updates can continue to appear here on OpenReview.